# High-Grade Endometrioid Stromal Sarcoma of the Ovary: Malignant Transformation of Ovarian Mature Cystic Teratoma

**DOI:** 10.3390/medicina58101501

**Published:** 2022-10-21

**Authors:** Hyoeun Kim, Jong Chul Baek

**Affiliations:** 1Department of Obstetrics and Gynecology, Gyeongsang National University Changwon Hospital, 11, Samjeongja-ro, Seongsan-gu, Changwon-si 51472, Korea; 2Department of Obstetrics and Gynecology, Gyeongsang National University School of Medicine, Jinju 52727, Korea; 3Institute of Health Science, Gyeongsang National University, Jinju 52727, Korea

**Keywords:** ovarian mature cystic teratoma, high-grade endometrioid stroma sarcoma, poor prognosis, malignant transformation

## Abstract

We report an extremely rare case of ovarian high-grade endometrioid stromal sarcoma arising from a mature cystic teratoma with clinicopathologic features, and then we briefly review the pertinent literature. A 62-year-old nulliparous woman presented with lower abdominal pain that had begun 6 months earlier. Magnetic resonance imaging showed two adnexal masses with fat components, which suggested that they were mature cystic teratomas. The eccentric thick rim of the left mass showed irregular invasion of the uterus, which was suggestive of malignancy. Positron emission tomography/computed tomography demonstrated high fluorodeoxyglucose uptake in the corresponding area. The patient underwent debulking cytoreductive surgery. The diagnosis was of an International Federation of Obstetrics and Gynecology stage IIIC high-grade endometrioid stromal sarcoma arising from a mature cystic teratoma. After surgery, the patient received adjuvant chemotherapy with three courses of doxorubicin regimen. The cancer recurred 3 months after surgery, and the patient died of progressive disease. It might be helpful for clinicians to be aware of this rare disease and the poor prognosis when it is at an advanced stage.

## 1. Introduction

Somatic neoplasms arising from teratomas are usually benign; in rare cases, malignant tumors may develop in the elements within a mature cystic teratoma (MCT) [1]. In such cases, the disease is known as “malignant transformation from ovarian mature cystic teratoma” (MT-MCT) [2,3]. Malignant transformation occurs in approximately 0.17~2% of MCTs; in approximately 80% of such cases, they become squamous cell carcinomas. Adenocarcinoma is the second most common carcinoma that develops from MT-MCTs (approximately 7% of cases). Other malignant sarcomas, including carcinosarcoma, angiosarcoma, and rhabdomyosarcoma, have also been reported [1,3,4,5,6]. The immunophenotype of MT-MCT is like that of malignancies of other organs at typical sites. Larger tumor size and elevated levels of tumor markers in patients older than 45 years are suggestive of malignant transformation [1,3,5]. The management of MT-MCT, which has not yet been standardized, often consists of cytoreductive surgery followed by adjuvant chemotherapy [7].

Endometrioid stromal sarcomas (ESSs) arise mainly from endometriosis. Their genetic alterations are similar to those in uterine endometrial stromal sarcomas. The majority of endometriosis-associated ovarian cancer was endometrioid or clear cell carcinomas [8]. Primary ovarian ESSs can be subcategorized as low-grade ESSs and high-grade ESSs (HG-ESS) [9]. HG-ESSs are extremely rare; to date, to the best of our knowledge, no description of malignant transformation from MCTs to HG-ESSs has yet been published, and no specific treatment has been recommended. Patients are treated in accordance with guidelines for uterine endometrial stromal sarcomas [10].

## 2. Case Report

A 62-year-old nulliparous woman presented with lower abdominal pain that had begun 6 months earlier. The patient had undergone menopause at 51 years of age. The patient was diagnosed with a dermoid cyst 20 years ago using pelvic ultrasonography, after which she had not undergone a follow-up pelvic USG examination. The patient had diabetes and no relevant surgical history. Serum tumor marker studies showed that the levels of squamous cell carcinoma antigen (SCC-Ag), alpha-fetoprotein (AFP), cancer antigen 125 (CA-125), CA 19-9, and carcinoembryonic antigen (CEA) were within normal ranges. On pelvic examination, a firm, fixed longitudinal mass without tenderness was detected.

Transvaginal ultrasonography revealed two cystic adnexal masses, one on each side, with echogenic nodules with distal acoustic shadowing; their widest dimensions were approximately 8.6 cm and 7.4 cm. Abdominal computer tomography (CT) depicted a fat-containing mass in each adnexa, and the left-sided mass had an eccentric thick rim (Figure 1A). Axial T1-weighted and contrast medium-enhanced T1-weighted magnetic resonance images (MRIs) showed fat components of both adnexal masses, which were suggestive of MCTs (Figure 1B,C). On sagittal T2-weighted images, the eccentric thick rim of the left-sided mass showed vivid contrast enhancement and irregular invasion of the uterus, which was suggestive of malignancy (Figure 1D). The mass was displacing the uterus anteriorly, and the endometrium of the atrophied uterus showed normal signal intensity (Figure 1E). In the left-sided fatty mass, enhancement of the solid portion at the periphery was suggestive of invasion of the uterine myometrium and left ureter. Left hydroureteronephrosis was evident, and no lymph nodes in the abdominopelvic cavity were significantly enlarged. Positron emission tomography/computed tomography (PET/CT) demonstrated high fluorodeoxyglucose uptake in the corresponding area of the eccentric thick rim of the left-sided mass but revealed no other distant metastases (Figure 1F). The clinical diagnosis was somatic malignancy arising from teratomas.

The patient underwent debulking cytoreductive surgery, including a total hysterectomy, salpingo-oophorectomy, omentectomy, and pelvic/para-aortic lymph node dissection. A solid nodule 1.2 cm in diameter in the omentum was identified, but no other metastatic site was visible in the abdominopelvic cavity.

Gross examination of the left-sided ovarian mass revealed cystic changes containing hair and necrotic fluid. The uterus itself was convex in shape, and the endometrium and myometrium were atrophied. Microscopic evaluation revealed that the left-sided mass contained mesenchymal cells of adipose tissue and epithelial components of skin (squamous epithelium, hair, and glands; Figure 2A). Hair was embedded in the cystic and sarcomatous elements of the mass (Figure 2B,C). The mass was composed of spindle-shaped cells arranged in intersecting fascicles (Figure 2D) and exhibited moderate pleomorphism and anaplastic change, with large nucleoli and severe necrosis. Mitosis was also observed, and the nuclei of the cells were hyperchromatic (Figure 2E). The left-sided mass showed dense cellularity and involved the ovary diffusely. Tumor cells were present in the outer portion of the left-sided myometrium. These tumor cells were strongly positive for CD10 and vimentin and were positive focally for cytokeratin and WT1 protein. They were negative for smooth muscle actin, desmin, and CD34 (Figure 2F). The histologic features of the resected ovarian specimens were similar to those of high-grade endometrial stromal sarcomas of the uterus. Pathologic study of the endometrium revealed atrophy. The omental nodule was confirmed to be a metastatic HG-ESS. The lymph nodes were free of disease. The final diagnosis was of an International Federation of Gynecology and Obstetrics (FIGO) stage IIIC HG-ESS arising in an MCT. After surgery, the patient received adjuvant chemotherapy with three courses of doxorubicin (75 mg/m^2^) every 3 weeks. After three cycles of adjuvant chemotherapy, whole abdominal ECT showed new lesions in the small bowel mesentery, left paracolic gutter, paraaortic L/N, and vaginal stump. The patient developed progressive disease and was treated with a palliative VIP (etoposide, ifosfamide, and cisplatin) regimen. However, the cancer recurred 3 months after surgery, and the patient died of exacerbation of the disease within 5 months after the diagnosis was confirmed.

## 3. Discussion

MCTs are germ cell tumors composed exclusively of mature tissues that develop from three germ cell layers (ectoderm, mesoderm, and endoderm) [1]. MCTs are among the most common ovarian tumors, representing 20–25% of all ovarian neoplasms [2]. MCTs are benign except in rare instances of concurrent malignant transformation. In rare cases, somatic malignancy develops in postmenopausal women [11]. Squamous cell carcinomas account for most cases of malignant transformation; adenocarcinomas and sarcomas account for a few [3,12]. Other benign and malignant ovarian tumors have been reported [7]. Immature teratomas differ from MCTs in that they have variable amounts of immature neuroectodermal tissue. MT-MCTs are common in postmenopausal women, while immature teratomas patients are often under 20 years of age. MT-MCTs must therefore be distinguished clinically from ovarian immature teratomas [3].

Most patients with MT-MCTs (median patient age is approximately 51 years, and the range is 21–75 years) are perimenopausal and postmenopausal, typically two decades older than those with benign MCTs [3]. Sarcomas occur more often in younger patients than do carcinomas. Most patients with MT-MCTs report abdominal pain at presentation; a minority report abdominal masses, bloating, change in bowel habits, and weight loss [3,5]. MT-MCT is thought to be a long-term complication of MCTs that remain in the pelvis [13]. No specific causes of MT-MCT are known. Until metastases have occurred, patients have no characteristic symptoms that would lead clinicians to suspect MT-MCT; thus, preoperative diagnosis is difficult. The risk of malignant transformation is significantly related to patient age (>45 years), tumor size (>10 cm), and elevated levels of tumor markers [1,3,5,7].

High serum levels of SCC-Ag, CA-125, cancer antigen 19-9, CEA, and AFP may be suggestive of malignant transformation. A high concentration of tumor markers was shown to be associated with a poor prognosis but not with tumor size [5,12]. The sensitivity and specificity of tumor markers for preoperative diagnosis are low [3]. In our patient, SCC-Ag, CA-125, CEA, and AFP levels were normal despite the advanced stage of disease; thus, normal ranges of tumor markers do not guarantee a good prognosis. The serum levels of tumor markers may be dependent on histologic subtype.

Previous reports of MT-MCT imaging have been limited. Some imaging characteristics on CT and MRI could indicate malignant transformation when invasive growth of an irregular border forms an obtuse angle with the teratoma wall, when a solid mass is larger than 5 cm in diameter, and when extracapsular growth extends into adjacent structures [14,15].

The macroscopic appearance of tumors is typically solid and cystic, with or without a recognizable dermoid cyst [1]. Dermoid plugs and Rokitansky nodules commonly accompany MCTs, but transmural invasive growth of the protuberance is conclusive evidence of MT-MCT [11,15]. Malignant transformation occurs in the dermoid plug; therefore, such plugs must be examined carefully, and any suspect or unusual portions of the cyst found during histopathologic investigation must be analyzed [11,14]. MT-MCTs should be distinguished from secondary involvement by a metastatic tumor [7]. Specimens from our patient had characteristics similar to those of high-grade endometrial stromal sarcomas of the uterus, as well as some characteristics of undifferentiated ESSs. The diagnostic criteria for uterine endometrial stromal sarcomas were not satisfied; thus, the tumor in this case was diagnosed as an HG-ESS resulting from MT-MCT.

ESSs are low-grade mesenchymal neoplasms with a structure resembling that of proliferative types of endometrial stroma [1]. Ovarian ESSs have genetic alterations similar to those of their uterine counterparts. Low-grade ESS typically have an indolent course, often recur later in life, and are much more common than HG-ESSs of the ovary [9]. The incidence of recurrence and metastasis with rapid progression of the disease is high for HG-ESSs, and they carry a poorer prognosis than do low-grade ESSs [9,16].

No specific treatment has been recommended for ovarian ESS [9]. Affected patients are usually treated in accordance with guidelines for uterine stromal sarcoma [10]. To date, ovarian HG-ESSs have not been classified as MT-MCTs; however, our case shows that ovarian teratomas can undergo malignant change in the absence of endometriosis. In primary ovarian sarcoma, older age and advanced FIGO stage are associated with a worse prognosis and a poorer oncologic outcome [17]. Our patient was 62 years old and had a FIGO stage IIIC HG-ESS of the ovary that recurred 3 months after surgery, and she died of progressive disease within 5 months after the diagnosis was confirmed. Although our patient represents a single case, it suggests that older age with advanced stage HG-ESS confers a poor prognosis.

Malignant changes of MCTs into squamous cell carcinoma should be suspected on the basis of the patient’s age, tumor size, imaging characteristics, and serum tumor markers [1,3,5]. The frequency of MT-MCT increases markedly after the age of 45 years [18]; the prognosis is better when a tumor’s capsule is intact; and a diameter greater than 10 cm and rapid growth of a tumor are associated with a high risk of malignant transformation [5,19]. The usefulness of tumor markers has been investigated in several studies, without consistent results. The SCC-Ag level appears to be the most relevant, especially in patients with recurrent lesions [11,20].

MT-MCTs are often managed with cytoreductive surgery followed by adjuvant chemotherapy [10]. However, MT-MCTs include many different histologic subtypes, and it would not make sense to treat the entire spectrum of the disease with the same treatment strategy. The current treatment for MT-MCT depends on the surgical stage and histologic features of the tumor. Although large studies might be challenging because of the rarity of the disease, standardization of treatment is needed. Laparotomy should be performed to prevent inadvertent spread of the tumor through spillages that can occur in laparoscopic surgery [21].

The prognosis of MT-MCTs depends strongly on stage, and most data are derived from cases of squamous cell carcinomas. With FIGO stage I tumors, the prognosis is favorable; the 5-year overall survival rate is 75–85%. In contrast, the 5-year overall survival rates for stages II and III tumors are less than 50% [5,20]. The prognosis for patients with advanced MT-MCTs is worse than that for patients with the more common epithelial ovarian malignancies.

## 4. Conclusions

MT-MCTs are rare gynecologic malignancies. Most information about these malignancies has been provided by descriptive series and reports of cases of squamous cell carcinoma. To better understand other tumors resulting from malignant transformations, such as adenocarcinoma and sarcoma, more detailed and specific studies of these rare entities are needed. To date, ovarian HG-ESSs have not been defined as MT-MCTs. The case of our patient shows that ovarian MCTs can undergo malignant transformation into HG-ESSs in the absence of endometriosis. The case of our patient suggests that older age and advanced stage HG-ESS confer a poor prognosis.

Careful close follow-up and the correct timing of surgery are necessary because the risk of malignant transformation is high in patients with MCTs who are older, have large tumors, or have elevated serum levels of tumor markers.

## Figures and Tables

**Figure 1 medicina-58-01501-f001:**
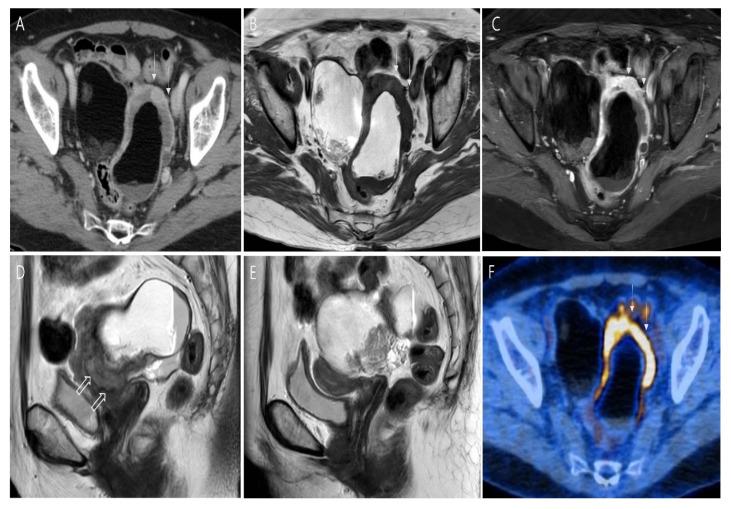
(**A**) Axial contrast medium-enhanced computed tomography showed a fat-containing mass in each adnexa. The left-sided mass had an eccentric thick rim (white arrows). Axial T1-weighted (**B**) and contrast medium-enhanced T1-weighted (**C**) magnetic resonance images (MRIs) showed suppression of high signal intensity of the fat component of both adnexal masses, which suggested that these masses were mature cystic teratomas. The eccentric thick rim of the left-sided mass ((**B**), arrows) showed vivid enhancement. (**D**) Sagittal T2-weighted MRI showed irregular invasion of the uterus (arrows), which indicated malignant transformation. (**E**) Sagittal T2-weighted MRI showed normal signal intensity of the endometrium of the atrophied uterus. (**F**) Positron emission tomography/computed tomography demonstrated high fluorodeoxyglucose uptake (arrows) in the corresponding area of the left-sided mass.

**Figure 2 medicina-58-01501-f002:**
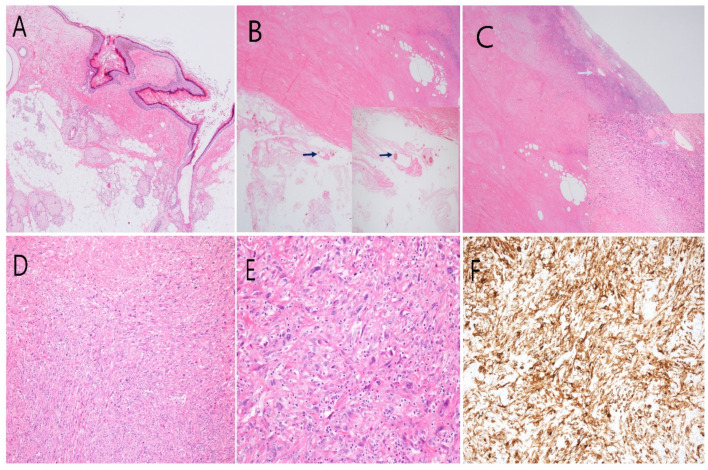
(**A**) Hematoxylin and eosin (H&E) staining of one specimen revealed that it was a teratoma, which contained mesenchymal cells of adipose tissue and epithelial components of skin (squamous epithelium, hair, and glands). (Original magnification, ×40). (**B**) Section of hair (arrows) embedded in the cystic component. (H&E stain; magnification, ×40). Inset: high-power view of the hair section. (H&E stain; magnification, ×100). (**C**) Section of sarcomatous component. (H&E stain; magnification, ×40). Inset: high-power view of sarcomatous component. (H&E stain; magnification, ×100). (**D**) Spindle-shaped cell proliferation with stromal pattern. (H&E stain; original magnification, ×100). (**E**) The tumors were composed of oval- to spindle-shaped cells and showed moderate pleomorphism and occasional anaplastic change, with large nucleoli. Mitosis was also observed. (H&E stain; original magnification, ×200). (**F**) Immunohistochemical analysis revealed strong positivity for CD10. (Original magnification, ×100).

## Data Availability

The datasets used and/or analyzed during the current study are available from the corresponding author upon reasonable request.

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
