# Peer review of "High-Grade Endometrioid Stromal Sarcoma of the Ovary: Malignant Transformation of Ovarian Mature Cystic Teratoma"

_medicina, 2022, doi:10.3390/medicina58101501_

Round 1

Reviewer 1 Report

How was the teratoma diagnosed, if the patient did not have ultrasound exam?

Please, explain how to differentiate ovarian inmature teratomas from MT-MCT

In  line 130  please  specify if data correspond to mature teratomas or MT-MCT.

 Could you explain in more detail the association of these ovarian masses and endometriosis?

 Ca 19.9 can be also elevated in teratomas. It is not commented in the text.

No information of transvaginal ultrasound results. As it is the first -level diagnostic tool,  ultrasould should be included , and also  ultrasound characteristics of teratoma, IOTA simple rules to evaluate adnexal masses. Paper of the doppler color.

Reviewer 2 Report

Dear Authors!

Congratulations on your article being the first to describe malignant transformation of mature cystic teratoma to endoemtrial stroma sarcoma in the ovary. The english of the article is clear, and the structure is well organized. I would have a few comments and specific question to the authors: 

Instead of reference No 10 please use the NCCN guidelines referring to high grade endometrial stroma sarcoma. 

In reference 9 and 10 there are former cases of ovarian endometrial stroma sarcoma described (primary ovarian). How do you prove this is not a co- existing condition of a former mature cystic teratoma (presenting for decades as you describe) and a newly starting ovarian endometrial stroma sarcoma of the same ovary? This would be a key question to answer, since this would make your article the first do describe a new entity.

Why did you use Doxorubicin + cysplatinum combination (in the NCCN guidelines this is not a preferred protocol). Why not mono doxorubicin or doxorubicin + ifosfamide combination?

You say the patient has received 6 courses q 3 weeks, however her tumour recurred 3 months after surgery. Her therapy should have lasted 15 months after the first cycle + time from surgery to first cycle (with pathology and tumour board approx 3 weeks). You also say she died 5 months after the diagnosis. Did you continue the chemotherapy despite progression? 

Did you check NTRK mutation, BRCA and MSI status for additional lines biological therapy? If no why not?
